# Multi-Task Learning for Contextual Bandits

**Aniket Anand Deshmukh**
Department of EECS
University of Michigan Ann Arbor
Ann Arbor, MI 48105
aniketde@umich.edu

**Urun Dogan**
Microsoft Research
Cambridge CB1 2FB, UK
urun.dogan@skype.net

**Clayton Scott**
Department of EECS
University of Michigan Ann Arbor
Ann Arbor, MI 48105
clayscot@umich.edu

## Abstract

Contextual bandits are a form of multi-armed bandit in which the agent has access
to predictive side information (known as the context) for each arm at each time step,
and have been used to model personalized news recommendation, ad placement,
and other applications. In this work, we propose a multi-task learning framework
for contextual bandit problems. Like multi-task learning in the batch setting, the
goal is to leverage similarities in contexts for different arms so as to improve the
agent's ability to predict rewards from contexts. We propose an upper confidence
bound-based multi-task learning algorithm for contextual bandits, establish a cor-
responding regret bound, and interpret this bound to quantify the advantages of
learning in the presence of high task (arm) similarity. We also describe an effective
scheme for estimating task similarity from data, and demonstrate our algorithm's
performance on several data sets.

## 1 Introduction

A multi-armed bandit (MAB) problem is a sequential decision making problem where, at each time
step, an agent chooses one of several "arms," and observes some reward for the choice it made. The
reward for each arm is random according to a fixed distribution, and the agent's goal is to maximize
its cumulative reward [4] through a combination of exploring different arms and exploiting those
arms that have yielded high rewards in the past [15, 11].

The contextual bandit problem is an extension of the MAB problem where there is some side
information, called the context, associated to each arm [12]. Each context determines the distribution
of rewards for the associated arm. The goal in contextual bandits is still to maximize the cumulative
reward, but now leveraging the contexts to predict the expected reward of each arm. Contextual
bandits have been employed to model various applications like news article recommendation [7],
computational advertisement [9], website optimization [20] and clinical trials [19]. For example, in
the case of news article recommendation, the agent must select a news article to recommend to a
particular user. The arms are articles and contextual features are features derived from the article and
the user. The reward is based on whether a user reads the recommended article.

One common approach to contextual bandits is to fix the class of policy functions (i.e., functions from
contexts to arms) and try to learn the best function with time [13, 18, 16]. Most algorithms estimate
rewards either separately for each arm, or have one single estimator that is applied to all arms. In

contrast, our approach is to adopt the perspective of multi-task learning (MTL). The intuition is that some arms may be similar to each other, in which case it should be possible to pool the historical data for these arms to estimate the mapping from context to rewards more rapidly. For example, in the case of news article recommendation, there may be thousands of articles, and some of those are bound to be similar to each other.

---

**Problem 1** Contextual Bandits

    **for** $t = 1, ..., T$ **do**
        Observe context $x_{a,t} \in \mathbb{R}^d$ for all arms $a \in [N]$, where $[N] = \{1, ...N\}$
        Choose an arm $a_t \in [N]$
        Receive a reward $r_{a_t,t} \in \mathbb{R}$
        Improve arm selection strategy based on new observation $(x_{a_t,t}, a_t, r_{a_t,t})$
    **end for**

---

The contextual bandit problem is formally stated in Problem 1. The total $T$ trial reward is defined as $\sum_{t=1}^{T} r_{a_t,t}$ and the optimal $T$ trial reward as $\sum_{t=1}^{T} r_{a_t^*,t}$, where $r_{a_t,t}$ is reward of the selected arm $a_t$ at time $t$ and $a_t^*$ is the arm with maximum reward at trial t. The goal is to find an algorithm that minimizes the $T$ trial regret

$$R(T) = \sum_{t=1}^{T} r_{a_t^*,t} - \sum_{t=1}^{T} r_{a_t,t}.$$

We focus on upper confidence bound (UCB) type algorithms for the remainder of the paper. A UCB strategy is a simple way to represent the exploration and exploitation tradeoff. For each arm, there is an upper bound on reward, comprised of two terms. The first term is a point estimate of the reward, and the second term reflects the confidence in the reward estimate. The strategy is to select the arm with maximum UCB. The second term dominates when the agent is not confident about its reward estimates, which promotes exploration. On the other hand, when all the confidence terms are small, the algorithm exploits the best arm(s) [2].

In the popular UCB type contextual bandits algorithm called Lin-UCB, the expected reward of an arm is modeled as a linear function of the context, $\mathbb{E}[r_{a,t}|x_{a,t}] = x_{a,t}^T \theta_a^*$, where $r_{a,t}$ is the reward of arm $a$ at time $t$ and $x_{a,t}$ is the context of arm $a$ at time $t$. To select the best arm, one estimates $\theta_a$ for each arm independently using the data for that particular arm [13]. In the language of multi-task learning, each arm is a task, and Lin-UCB learns each task independently.

In the theoretical analysis of the Lin-UCB [7] and its kernelized version Kernel-UCB [18] $\theta_a$ is replaced by $\theta$, and the goal is to learn one single estimator using data from all the arms. In other words, the data from the different arms are pooled together and viewed as coming from a single task. These two approaches, independent and pooled learning, are two extremes, and reality often lies somewhere in between. In the MTL approach, we seek to pool some tasks together, while learning others independently.

We present an algorithm motivated by this idea and call it kernelized multi-task learning UCB (KMTL-UCB). Our main contributions are proposing a UCB type multi-task learning algorithm for contextual bandits, established a regret bound and interpreting the bound to reveal the impact of increased task similarity, introducing a technique for estimating task similarities on the fly, and demonstrating the effectiveness of our algorithm on several datasets.

This paper is organized as follows. Section 2 describes related work and in Section 3 we propose a UCB algorithm using multi-task learning. Regret analysis is presented in Section 4, and our experimental findings are reported in Section 5. We conclude in Section 6.

## 2 Related Work

A UCB strategy is a common approach to quantify the exploration/exploitation tradeoff. At each time step $t$, and for each arm $a$, a UCB strategy estimates a reward $\hat{r}_{a,t}$ and a one-sided confidence interval above $\hat{r}_{a,t}$ with width $\hat{w}_{a,t}$. The term $ucb_{a,t} = \hat{r}_{a,t} + \hat{w}_{a,t}$ is called the UCB index or just UCB. Then at each time step $t$, the algorithm chooses the arm $a$ with the highest UCB.

In contextual bandits, the idea is to view learning the mapping $x \mapsto r$ as a regression problem. Lin-UCB uses a linear regression model while Kernel-UCB uses a nonlinear regression model drawn from the reproducing kernel Hilbert space (RKHS) of a symmetric and positive definite (SPD) kernel. Either of these two regression models could be applied in either the *independent* setting or the *pooled* setting. In the independent setting, the regression function for each arm is estimated separately. This was the approach adopted by Li et al. [13] with a linear model. Regret analysis for both Lin-UCB and Kernel-UCB adopted the *pooled* setting [7, 18]. Kernel-UCB in the independent setting has not previously been considered to our knowledge, although the algorithm would just be a kernelized version of Li et al. [13]. We will propose a methodology that extends the above four combinations of setting (independent and pooled) and regression model (linear and nonlinear). Gaussian Process UCB (GP-UCB) uses a Gaussian prior on the regression function and is a Bayesian equivalent of Kernel-UCB [16].

There are some contextual bandit setups that incorporate multi-task learning. In Lin-UCB with Hybrid Linear Models the estimated reward consists of two linear terms, one that is arm-specific and another that is common to all arms [13]. Gang of bandits [5] uses a graph structure (e.g., a social network) to transfer the learning from one user to other for personalized recommendation. Collaborative filtering bandits [14] is a similar technique which clusters the users based on context. Contextual Gaussian Process UCB (CGP-UCB) builds on GP-UCB and has many elements in common with our framework [10]. We defer a more detailed comparison to CGP-UCB until later.

## 3   KMTL-UCB

We propose an alternate regression model that includes the independent and pooled settings as special cases. Our approach is inspired by work on transfer and multi-task learning in the batch setting [3, 8]. Intuitively, if two arms (tasks) are similar, we can pool the data for those arms to train better predictors for both.

Formally, we consider regression functions of the form

$$f : \tilde{X} \mapsto \mathcal{Y}$$

where $\tilde{X} = \mathcal{Z} \times \mathcal{X}$, and $\mathcal{Z}$ is what we call the *task similarity space*, $\mathcal{X}$ is the *context space* and $\mathcal{Y} \subseteq \mathbb{R}$ is the reward space. Every context $x_a \in \mathcal{X}$ is associated with an arm descriptor $z_a \in \mathcal{Z}$, and we define $\tilde{x}_a = (z_a, x_a)$ to be the *augmented context*. Intuitively, $z_a$ is a variable that can be used to determine the similarity between different arms. Examples of $\mathcal{Z}$ and $z_a$ will be given below.

Let $\tilde{k}$ be a SPD kernel on $\tilde{X}$. In this work we focus on kernels of the form

$$\tilde{k}\big((z, x), (z', x')\big) = k_{\mathcal{Z}}(z, z') k_{\mathcal{X}}(x, x'), \tag{1}$$

where $k_{\mathcal{X}}$ is a SPD kernel on $\mathcal{X}$, such as linear or Gaussian kernel if $\mathcal{X} = \mathbb{R}^d$, and $k_{\mathcal{Z}}$ is a kernel on $\mathcal{Z}$ (examples given below). Let $\mathcal{H}_{\tilde{k}}$ be the RKHS of functions $f : \tilde{X} \mapsto \mathbb{R}$ associated to $\tilde{k}$. Note that a product kernel is just one option for $\tilde{k}$, and other forms may be worth exploring.

### 3.1   Upper Confidence Bound

Instead of learning regression estimates for each arm separately, we effectively learn regression estimates for all arms at once by using all the available training data. Let $N$ be the total number of distinct arms that algorithm has to choose from. Define $[N] = \{1, ..., N\}$ and let the observed contexts at time $t$ be $x_{a,t}, \forall a \in [N]$. Let $n_{a,t}$ be the number of times the algorithm has selected arm $a$ up to and including time $t$ so that $\sum_{a=1}^{N} n_{a,t} = t$. Define sets $t_a = \{\tau < t : a_\tau = a\}$, where $a_\tau$ is the arm selected at time $\tau$. Notice that $|t_a| = n_{a,t-1}$ for all $a$. We solve the following problem at time $t$:

$$\hat{f}_t = \operatorname*{arg\,min}_{f \in \mathcal{H}_{\tilde{k}}} \frac{1}{N} \sum_{a=1}^{N} \frac{1}{n_{a,t-1}} \sum_{\tau \in t_a} (f(\tilde{x}_{a,\tau}) - r_{a,\tau})^2 + \lambda \|f\|_{\mathcal{H}_{\tilde{k}}}^2, \tag{2}$$

where $\tilde{x}_{a,\tau}$ is the augmented context of arm $a$ at time $\tau$, and $r_{a,\tau}$ is the reward of an arm $a$ selected at time $\tau$. This problem (2) is a variant of kernel ridge regression. Applying the representer theorem [17]

the optimal $f$ can be expressed as $f = \sum_{a'=1}^{N} \sum_{\tau' \in t_{a'}} \alpha_{a',\tau'} \tilde{k}(\cdot, \tilde{x}_{a',\tau'})$, which yields the solution (detailed derivation is in the supplementary material)

$$\hat{f}_t(\tilde{x}) = \tilde{k}_{t-1}(\tilde{x})^T (\eta_{t-1} \tilde{K}_{t-1} + \lambda I)^{-1} \eta_{t-1} y_{t-1}, \tag{3}$$

where $\tilde{K}_{t-1}$ is the $(t-1) \times (t-1)$ kernel matrix on the augmented data $[\tilde{x}_{a_\tau,\tau}]_{\tau=1}^{t-1}$, $\tilde{k}_{t-1}(\tilde{x}) = [\tilde{k}(\tilde{x}, \tilde{x}_{a_\tau,\tau})]_{\tau=1}^{t-1}$ is a vector of kernel evaluations between $\tilde{x}$ and the past data, $y_{t-1} = [r_{a_\tau,\tau}]_{\tau=1}^{t-1}$ are all observed rewards, and $\eta_{t-1}$ is the $(t-1) \times (t-1)$ diagonal matrix $\eta_{t-1} = \text{diag}[\frac{1}{n_{a_\tau,t-1}}]_{\tau=1}^{t-1}$.

When $\tilde{x} = \tilde{x}_{a,t}$, we write $\tilde{k}_{a,t} = \tilde{k}_{t-1}(\tilde{x}_{a,t})$. With only minor modifications to the argument in Valko et al [18], we have the following:

**Lemma 1.** *Suppose the rewards $[r_{a_\tau,\tau}]_{\tau=1}^{T}$ are independent random variables with means $\mathbb{E}[r_{a_\tau,\tau}|\tilde{x}_{a_\tau,\tau}] = f^*(\tilde{x}_{a_\tau,\tau})$, where $f^* \in \mathcal{H}_{\tilde{k}}$ and $\|f^*\|_{\mathcal{H}_{\tilde{k}}} \leq c$. Let $\alpha = \sqrt{\frac{\log(2TN/\delta)}{2}}$ and $\delta > 0$. With probability at least $1 - \frac{\delta}{T}$, we have that $\forall a \in [N]$*

$$|\hat{f}_t(\tilde{x}_{a,t}) - f^*(\tilde{x}_{a,t})| \leq w_{a,t} := (\alpha + c\sqrt{\lambda})s_{a,t} \tag{4}$$

*where $s_{a,t} = \lambda^{-1/2}\sqrt{\tilde{k}(\tilde{x}_{a,t}, \tilde{x}_{a,t}) - \tilde{k}_{a,t}^T (\eta_{t-1} \tilde{K}_{t-1} + \lambda I)^{-1} \eta_{t-1} \tilde{k}_{a,t}}$.*

The result in Lemma 1 motivates the UCB

$$ucb_{a,t} = \hat{f}_t(\tilde{x}_{a,t}) + w_{a,t}$$

and inspires Algorithm 1.

---

**Algorithm 1** KMTL-UCB

> **Input:** $\beta \in R_+$,
> **for** $t = 1, ..., T$ **do**
>> Update the (product) kernel matrix $\tilde{K}_{t-1}$ and $\eta_{t-1}$
>> Observe context features at time $t$: $x_{a,t}$ for each $a \in [N]$.
>> Determine arm descriptor $z_a$ for each $a \in [N]$ to get augmented context $\tilde{x}_{a,t}$.
>> **for** all $a$ at time $t$ **do**
>>> $p_{a,t} \leftarrow \hat{f}_t(\tilde{x}_{a,t}) + \beta s_{a,t}$
>> **end for**
>> Choose arm $a_t = \arg\max p_{a,t}$, observe a real valued payoff $r_{a_t,t}$ and update $y_t$ .
>> **Output:** $a_t$
> **end for**

---

Before an arm has been selected at least once, $\hat{f}_t(\tilde{x}_{a,t})$ and the second term in $s_{a,t}$, i.e., $\tilde{k}_{a,t}^T(\eta_{t-1}\tilde{K}_{t-1} + \lambda I)^{-1}\eta_{t-1}\tilde{k}_{a,t}$, are taken to be 0. In that case, the algorithm only uses the first term of $s_{a,t}$, i.e., $\sqrt{\tilde{k}(\tilde{x}_{a,t}, \tilde{x}_{a,t})}$, to form the UCB.

## 3.2 Choice of Task Similarity Space and Kernel

To illustrate the flexibility of our framework, we present the following three options for $\mathcal{Z}$ and $k_{\mathcal{Z}}$:

1. Independent: $\mathcal{Z} = \{1, ..., N\}$, $k_{\mathcal{Z}}(a, a') = \mathbb{1}_{a=a'}$. The augmented context for a context $x_a$ from arm $a$ is just $(a, x_a)$.

2. Pooled: $\mathcal{Z} = \{1\}$, $k_{\mathcal{Z}} \equiv 1$. The augmented context for a context $x_a$ for arm $a$ is just $(1, x_a)$.

3. Multi-Task: $\mathcal{Z} = \{1, ..., N\}$ and $k_{\mathcal{Z}}$ is a PSD matrix reflecting arm/task similarities. If this matrix is unknown, it can be estimated as discussed below.

Algorithm 1 with the first two choices specializes to the independent and pooled settings mentioned previously. In either setting, choosing a linear kernel for $k_{\mathcal{X}}$ leads to Lin-UCB, while a more general kernel essentially gives rise to Kernel-UCB. We will argue that the multi-task setting facilitates learning when there is high task similarity.

We also introduce a fourth option for $\mathcal{Z}$ and $k_{\mathcal{Z}}$ that allows task similarity to be estimated when it is unknown. In particular, we are inspired by the kernel transfer learning framework of Blanchard et al. [3]. Thus, we define the arm similarity space to be $\mathcal{Z} = \mathcal{P}_{\mathcal{X}}$, the set of all probability distributions on $\mathcal{X}$. We further assume that contexts for arm $a$ are drawn from probability measure $P_a$. Given a context $x_a$ for arm $a$, we define its augmented context to be $(P_a, x_a)$.

To define a kernel on $\mathcal{Z} = \mathcal{P}_{\mathcal{X}}$, we use the same construction described in [3], originally introduced by Steinwart and Christmann [6]. In particular, in our experiments we use a Gaussian-like kernel

$$k_{\mathcal{Z}}(P_a, P_{a'}) = \exp(-\|\Psi(P_a) - \Psi(P_{a'})\|^2 / 2\sigma_{\mathcal{Z}}^2), \tag{5}$$

where $\Psi(P) = \int k'_{\mathcal{X}}(\cdot, x) dPx$ is the kernel mean embedding of a distribution $P$. This embedding is defined by yet another SPD kernel $k'_{\mathcal{X}}$ on $\mathcal{X}$, which could be different from the $k_{\mathcal{X}}$ used to define $\tilde{k}$. We may estimate $\Psi(P_a)$ via $\Psi(\widehat{P}_a) = \frac{1}{n_{a,t-1}} \sum_{\tau \in t_a} k'_{\mathcal{X}}(\cdot, x_{a_\tau, \tau})$, which leads to an estimate of $k_{\mathcal{Z}}$.

# 4 Theoretical Analysis

To simplify the analysis we consider a modified version of the original problem 2:

$$\hat{f}_t = \arg\min_{f \in \mathcal{H}_{\tilde{k}}} \frac{1}{N} \sum_{a=1}^{N} \sum_{\tau \in t_a} (f(\tilde{x}_{a,\tau}) - r_{a,\tau})^2 + \lambda \|f\|_{\mathcal{H}_{\tilde{k}}}^2. \tag{6}$$

In particular, this modified problem omits the terms $\frac{1}{n_{a,t-1}}$ as they obscure the analysis. In practice, these terms should be incorporated.

In this case $s_{a,t} = \lambda^{-1/2} \sqrt{\tilde{k}(\tilde{x}_{a,t}, \tilde{x}_{a,t}) - \tilde{k}_{a,t}^T (\tilde{K}_{t-1} + \lambda I)^{-1} \tilde{k}_{a,t}}$. Under this assumption Kernel-UCB is exactly KMTL-UCB with $k_{\mathcal{Z}} \equiv 1$. On the other hand, KMTL-UCB can be viewed as a special case of Kernel-UCB on the augmented context space $\tilde{\mathcal{X}}$. Thus, the regret analysis of Kernel-UCB applies to KMTL-UCB, but it does not reveal the potential gains of multi-task learning. We present an interpretable regret bound that reveals the benefits of MTL. We also establish a lower bound on the UCB width that decreases as task similarity increases (presented in the supplementary file).

## 4.1 Analysis of SupKMTL-UCB

It is not trivial to analyze algorithm 1 because the reward at time $t$ is dependent on the past rewards. We follow the same strategy originally proposed in [1] and used in [7, 18] which uses SupKMTL-UCB as a master algorithm, and BaseKMTL-UCB (which is called by SupKMTL-UCB) to get estimates of reward and width. SupKMTL-UCB builds mutually exclusive subsets of $[T]$ such that rewards in any subset are independent. This guarantees that the independence assumption of Lemma 1 is satisfied. We describe these algorithms in a supplementary section because of space constraints.

**Theorem 1.** *Assume that $r_{a,t} \in [0, 1], \forall a \in [N]$, $T \geq 1$, $\|f^*\|_{\mathcal{H}_{\tilde{k}}} \leq c$, $\tilde{k}(\tilde{x}, \tilde{x}) \leq c_{\tilde{k}}, \forall \tilde{x} \in \tilde{X}$ and the task similarity matrix $K_Z$ is known. With probability at least $1 - \delta$, SupKMTL-UCB satisfies*

$$
\begin{aligned}
R(T) &\leq 2\sqrt{T} + 10\left(\sqrt{\frac{\log\left(2TN(\log(T)+1)/\delta\right)}{2}} + c\sqrt{\lambda}\right) \sqrt{2m \log g([T])} \sqrt{T\lceil \log(T)\rceil} \\
&= O\left(\sqrt{T \log(g([T]))}\right)
\end{aligned}
$$

*where $g([T]) = \frac{\det(\tilde{K}_{T+1} + \lambda I)}{\lambda^{T+1}}$ and $m = \max(1, \frac{c_{\tilde{k}}}{\lambda})$.*

Note that this theorem assumes that task similarity is known. In the experiments for real datasets using the approach discussed in subsection 3.2 we estimate the task similarity from the available data.

## 4.2 Interpretation of Regret Bound

The following theorems help us interpret the regret bound by looking at

$$g([T]) = \frac{\det(\tilde{K}_{T+1} + \lambda I)}{\lambda^{T+1}} = \prod_{t=1}^{T+1} \frac{(\lambda_t + \lambda)}{\lambda},$$

where, $\lambda_1 \geq \lambda_2 \geq \cdots \geq \lambda_{T+1}$ are the eigenvalues of the kernel matrix $\tilde{K}_{T+1}$.

As mentioned above, the regret bound of Kernel-UCB applies to our method, and we are able to recover this bound as a corollary of Theorem 1. In the case of Kernel-UCB $\tilde{K}_t = K_{X_t}, \forall t \in [T]$ as all arm estimators are assumed to be the same. We define the effective rank of $\tilde{K}_{T+1}$ in the same way as [18] defines the effective dimension of the kernel feature space.

**Definition 1.** *The effective rank of $\tilde{K}_{T+1}$ is defined to be* $r := \min\{j : j\lambda \log T \geq \sum_{i=j+1}^{T+1} \lambda_i\}$.

In the following result, the notation $\tilde{O}$ hides logarithmic terms.

**Corollary 1.** $\log(g([T])) \leq r \log \left( 2T \frac{2(T+1)c_{\tilde{k}}+r\lambda-r\lambda\log T}{r\lambda} \right)$, and therefore $R(T) = \tilde{O}(\sqrt{rT})$

However, beyond recovering a known bound, Theorem 1 can also be interpreted to reveal the potential gains of multi-task learning. To interpret the regret bound in Theorem 1, we make a further assumption that after time $t$, $n_{a,t} = \frac{t}{N}$ for all $a \in [N]$. For simplicity define $n_t = n_{a,t}$. Let $\odot$ denote the Hadamard product, $\otimes$ denote the Kronecker product and $\mathbb{1}_n \in R^n$ be the vector of ones. Let $K_{X_t} = [k_{\mathcal{X}}(x_{a_\tau,\tau}, x_{a_{\tau'},\tau'})]_{\tau,\tau'=1}^t$ be the $t \times t$ kernel matrix on contexts, $K_{Z_t} = [k_{\mathcal{Z}}(z_{a_\tau}, z_{a_{\tau'}})]_{\tau,\tau'=1}^t$ be the associated $t \times t$ kernel matrix based on arm similarity, and $K_Z = [k_{\mathcal{Z}}(z_a, z_a)]_{a=1}^N$ be the $N \times N$ arm/task similarity matrix between N arms, where $x_{a_\tau,\tau}$ is the observed context and $z_{a_\tau}$ is the associated arm descriptor. Using eqn. (1), we can write $\tilde{K}_t = K_{Z_t} \odot K_{X_t}$. We rearrange the sequence of $x_{a_\tau,\tau}$ to get $[x_{a,\tau}]_{a=1,\tau=(t+1)_a}^N$ such that elements $(a-1)n_t$ to $an_t$ belong to arm $a$. Define $\tilde{K}_t^r, K_{X_t}^r$ and $K_{Z_t}^r$ to be the rearranged kernel matrices based on the re-ordered set $[x_{a,\tau}]_{a=1,\tau=(t+1)_a}^N$. Notice that we can write $\tilde{K}_t^r = (K_Z \otimes \mathbb{1}_{n_t}\mathbb{1}_{n_t}^T) \odot K_{X_t}^r$ and the eigenvalues $\lambda(\tilde{K}_t)$ and $\lambda(\tilde{K}_t^r)$ are equal. To summarize, we have

$$\tilde{K}_t = K_{Z_t} \odot K_{X_t}$$
$$\lambda(\tilde{K}_t) = \lambda\left((K_Z \otimes \mathbb{1}_{n_t}\mathbb{1}_{n_t}^T) \odot K_{X_t}^r\right). \tag{7}$$

**Theorem 2.** *Let the rank of matrix $K_{X_{T+1}}$ be $r_x$ and the rank of matrix $K_Z$ be $r_z$. Then* $\log(g([T])) \leq r_z r_x \log \left( \frac{(T+1)c_{\tilde{k}}+\lambda}{\lambda} \right)$

This means that when the rank of the task similarity matrix is low, which reflects a high degree of inter-task similarity, the regret bound is tighter. For comparison, note that when all tasks are independent, $r_z = N$ and when all tasks are the same (pooled), then $r_z = 1$. In the case of Lin-UCB [7] where all arm estimators are assumed to be the same and $k_{\mathcal{X}}$ is a linear kernel, the regret bound in Theorem 1 evaluates to $\tilde{O}(\sqrt{dT})$, where $d$ is the dimension of the context space. In the original Lin-UCB algorithm [13] where all arm estimators are different, the regret bound would be $\tilde{O}(\sqrt{NdT})$.

We can further comment on $g([T])$ when all distinct tasks (arms) are similar to each other with task similarity equal to $\mu$. Thus define $K_Z(\mu) := (1-\mu)I_N + \mu\mathbb{1}_N\mathbb{1}_N^T$ and $\tilde{K}_t^r(\mu) = (K_Z(\mu) \otimes \mathbb{1}_{n_t}\mathbb{1}_{n_t}^T) \odot K_{X_t}^r$.

**Theorem 3.** *Let $g_\mu([T]) = \frac{\det(\tilde{K}_{T+1}^r(\mu)+\lambda I)}{\lambda^{T+1}}$. If $\mu_1 \leq \mu_2$ then $g_{\mu_1}([T]) \geq g_{\mu_2}([T])$.*

This shows that when there is more task similarity, the regret bound is tighter.

### 4.3 Comparison with CGP-UCB

CGP-UCB transfers the learning from one task to another by leveraging additional known task-specific context variables [10], similar in spirit to KTML-UCB. Indeed, with slight modifications, KMTL-UCB can be viewed as a frequentist analogue of CGP-UCB, and similarly CGP-UCB could be modified to address our setting. Furthermore, the term $g([T])$ appearing in our regret bound is equivalent to an information gain term used to analyze CGP-UCB. In the agnostic case of CGP-UCB where there is no assumption of a Gaussian prior on decision functions, their regret bound is $O(\log(g([T]))\sqrt{T})$, while their regret bound matches ours when they adopt a GP prior on $f^*$. Thus, our primary contributions with respect to CGP-UCB are to provide a tighter regret bound in agnostic case, and a technique for estimating task similarity which is critical for real-world applications.

# 5 Experiments

We test our algorithm on synthetic data and some multi-class classification datasets. In the case of multi-class datasets, the number of arms $N$ is the number of classes and the reward is $1$ if we predict the correct class, otherwise it is $0$. We separate the data into two parts - validation set and test set. We use all Gaussian kernels and pre-select the bandwidth of kernels using five fold cross-validation on a holdout validation set and we use $\beta = 0.1$ for all experiments. Then we run the algorithm on the test set 10 times (with different sequences of streaming data) and report the mean regret. For the synthetic data, we compare Kernel-UCB in the independent setting (Kernel-UCB-Ind) and pooled setting (Kernel-UCB-Pool), KMTL-UCB with known task similarity, and KMTL-UCB-Est which estimates task similarity on the fly. For the real datasets in the multi-class classification setting, we compare Kernel-UCB-Ind and KMTL-UCB-Est. In this case, the pooled setting is not valid because $x_{a,t}$ is the same for all arms (only $z_a$ differs) and KMTL-UCB is not valid because the task similarity matrix is unknown. We also report the confidence intervals for these results in the supplementary material.

## 5.1 Synthetic News Article Data

Suppose an agent has access to a pool of articles and their context features. The agent then sees a user along with his/her features for which it needs to recommend an article. Based on user features and article features the algorithm gets a combined context $x_{a,t}$. The user context $x_{u,t} \in \mathbb{R}^2, \forall t$ is randomly drawn from an ellipse centered at $(0, 0)$ with major axis length 1 and minor axis length 0.5. Let $x_{u,t}[:, 1]$ be the minor axis and $x_{u,t}[:, 2]$ be the major axis. Article context $x_{art,t}$ is any angle $\theta \in [0, \frac{\pi}{2}]$. To get the overall summary $x_{a,t}$ of user and article the user context $x_{u,t}$ is rotated with $x_{art,t}$. Rewards for each article are defined based on the minor axis $r_{a,t} = \left(1.0 - (x_{u,t}[:, 1] - \frac{a}{N} + 0.5)^2\right)$.

Figure 1: Synthetic Data

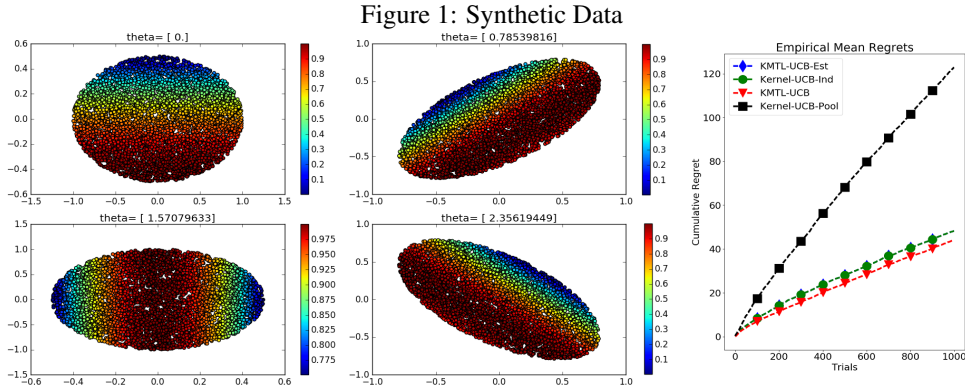

Figure 1 shows one such example for $4$ different arms. The color code describes the reward, the two axes show the information about user context, and theta is the article context. We take $N = 5$. For KMTL-UCB, we use a Gaussian kernel on $x_{art,t}$ to get the task similarity.

The results of this experiment are shown in Figure 1. As one can see, Kernel-UCB-Pool performs the worst. That means for this setting combining all the data and learning a single estimator is not efficient. KMTL-UCB beats the other methods in all 10 runs, and Kernel-UCB-Ind and KMTL-UCB-Est perform equally well.

## 5.2 Multi-class Datasets

In the case of multi-class classification, each class is an arm and the features of an example for which the algorithm needs to recommend a class are the contexts. We consider the following datasets: Digits ($N = 10, d = 64$), Letter ($N = 26, d = 16$), MNIST ($N = 10, d = 780$), Pendigits ($N = 10, d = 16$), Segment ($N = 7, d = 19$) and USPS ($N = 10, d = 256$). Empirical mean regrets are shown in Figure 2. KMTL-UCB-Est performs the best in three of the datasets and performs equally well in the other three datasets. Figure 3 shows the estimated task similarity (re-ordered

to reveal block structure) and one can see the effect of the estimated task similarity matrix on the empirical regret in Figure 2. For the Digits, Segment and MNIST datasets, there is significant inter-task similarity. For Digits and Segment datasets, KMTL-UCB-Est is the best in all 10 runs of the experiment while for MNIST, KMTL-UCB-Est is better for all but 1 run.

Figure 2: Results on Multiclass Datasets - Empirical Mean Regret

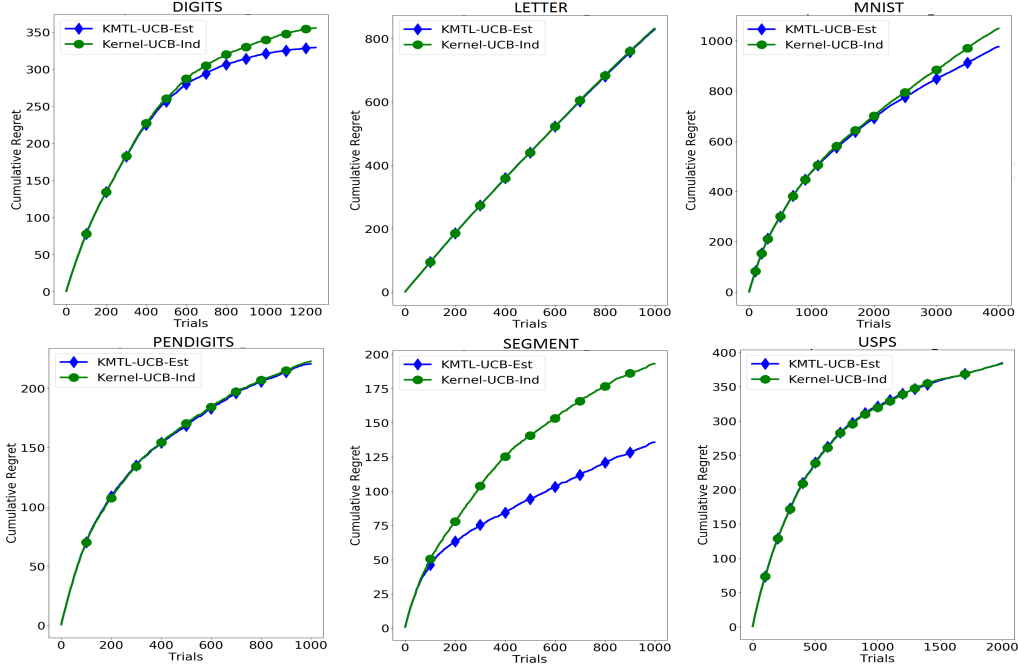

Figure 3: Estimated Task Similarity for Real Datasets

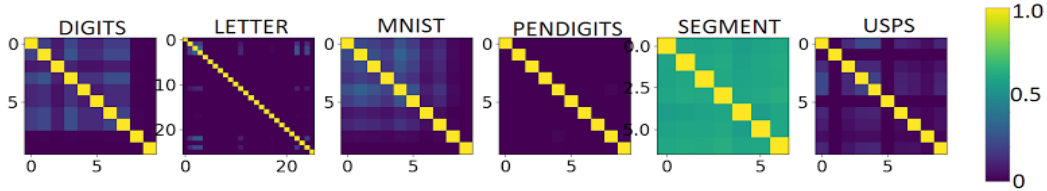

# 6 Conclusions and future work

We present a multi-task learning framework in the contextual bandit setting and describe a way to estimate task similarity when it is not given. We give theoretical analysis, interpret the regret bound, and support the theoretical analysis with extensive experiments. In the supplementary material we establish a lower bound on the UCB width, and argue that it decreases as task similarity increases.

Our proposal to estimate the task similarity matrix using the arm similarity space $\mathcal{Z} = P_{\mathcal{X}}$ can be extended in different ways. For example, we could also incorporate previously observed rewards into $\mathcal{Z}$. This would alleviate a potential problem with our approach, namely, that some contexts may have been selected when they did not yield a high reward. Additionally, by estimating the task similarity matrix, we are estimating arm-specific information. In the case of multiclass classification, $k_{\mathcal{Z}}$ reflects information that represents the various classes. A natural extension is to incorporate methods for representation learning into the MTL bandit setting.

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
