[Supplementary Material · supplementaty.pdf]

# Multi Task Learning for Contextual Bandits - Supplementary Material

**Aniket Anand Deshmukh**
Department of EECS
University of Michigan Ann Arbor
Ann Arbor, MI 48105
`aniketde@umich.edu`

**Urun Dogan**
Microsoft Research
Cambridge CB1 2FB, UK
`urun.dogan@skype.net`

**Clayton Scott**
Department of EECS
University of Michigan Ann Arbor
Ann Arbor, MI 48105
`clayscot@umich.edu`

## 1   KMTL Ridge Regression

Let $n_{a,t}$ be the number of times the algorithm has selected arm $a$ up and including time $t$ so that $\sum_{a=1}^{N} n_{a,t} = t$. Define sets $t_a = \{\tau < t : a_\tau = a\}$, where $a_\tau$ is the arm selected at time $\tau$. Notice that $|t_a| = n_{a,t-1}$ for all $a$. We solve the following problem at time $t$:

$$\hat{f}_t = \arg\min_{f \in \mathcal{H}_{\tilde{k}}} \frac{1}{N} \sum_{a=1}^{N} \frac{1}{n_{a,t-1}} \sum_{\tau \in t_a} (f(\tilde{x}_{a,\tau}) - r_{a,\tau})^2 + \lambda \|f\|_{\mathcal{H}_{\tilde{k}}}^2, \tag{1}$$

where $\tilde{x}_{a,\tau}$ is augmented context and $r_{a,\tau}$ is the reward of arm $a$ selected at time $\tau$. We can minimize (1) by solving a variant of kernel ridge regression. Applying the representer theorem [10] the optimal $f$ can be expressed as $f = \sum_{a'=1}^{N} \sum_{\tau' \in t_a} \alpha_{a'\tau'} \tilde{k}(\cdot, \tilde{x}_{a',\tau'})$. Plugging this in, we have the objective function

$$
\begin{aligned}
J(f) &= \frac{1}{N} \sum_{a=1}^{N} \frac{1}{n_{a,t-1}} \sum_{\tau \in t_a} (\sum_{a'=1}^{N} \sum_{\tau' \in t_a} \alpha_{a'\tau'} \tilde{k}(\tilde{x}_{a,\tau}, \tilde{x}_{a',\tau'}) - r_{a,\tau})^2 + \lambda \|f\|_{\mathcal{H}_{\tilde{k}}}^2 \\
&= (y_{t-1} - \tilde{K}_{t-1}\alpha)^T \eta_{t-1}(y_{t-1} - \tilde{K}_{t-1}\alpha) + \lambda \alpha^T \tilde{K}_{t-1}\alpha \\
&= y_{t-1}^T \eta_{t-1} y_{t-1} - y_{t-1}^T \eta_{t-1} \tilde{K}_{t-1}\alpha - \alpha^T \tilde{K}_{t-1}\eta_{t-1} y_{t-1} \\
&\quad + \alpha^T \tilde{K}_{t-1}\eta_{t-1}\tilde{K}_{t-1}\alpha + \lambda \alpha^T \tilde{K}_{t-1}\alpha.
\end{aligned}
$$

Taking the gradient, we have

$$\frac{\partial J}{\partial \alpha} = -2\tilde{K}_{t-1}\eta_{t-1} y_{t-1} + 2\tilde{K}_{t-1}\eta_{t-1}\tilde{K}_{t-1}\alpha + 2\lambda \tilde{K}_{t-1}\alpha = 0.$$

Solving for $\alpha$ yields

$$\alpha = (\eta_{t-1}\tilde{K}_{t-1} + \lambda I)^{-1}\eta_{t-1} y_{t-1},$$

which implies

$$\hat{f}_t(\tilde{x}) = \tilde{k}_{t-1}(\tilde{x})^T (\eta_{t-1} \tilde{K}_{t-1} + \lambda I)^{-1} \eta_{t-1} y_{t-1}. \tag{2}$$

Here $\tilde{K}_{t-1}$ is the $(t-1) \times (t-1)$ kernel matrix on the augmented data $[\tilde{x}_{a_\tau,\tau}]_{\tau=1}^{t-1}$, $\tilde{k}_{t-1}(\tilde{x}) = [\tilde{k}(\tilde{x}, \tilde{x}_{a_\tau,\tau})]_{\tau=1}^{t-1}$ is a vector of kernel evaluations between $\tilde{x}$ and the past data, $y_{t-1} = [r_{a_\tau,\tau}]_{\tau=1}^{t-1}$ are all observed labels or rewards and $\eta_{t-1}$ is the $(t-1) \times (t-1)$ diagonal matrix $\eta_{t-1} = \text{diag}[\frac{1}{n_{a_\tau}}]_{\tau=1}^{t-1}$.

We can also derive the solution without using the representer theorem. Let $\phi$ be a feature map associated with kernel $\tilde{k}$. Let

$$\hat{\theta} = \arg \min_\theta \frac{1}{N} \sum_{a=1}^{N} \frac{1}{n_{a,t-1}} \sum_{\tau \in t_a} (\phi(\tilde{x}_{a,\tau})^T \theta - r_{a,\tau})^2 + \lambda \|\theta\|^2. \tag{3}$$

Minimizing eqn. (3) over $\theta$ gives,

$$\hat{\theta}_t = D_{t-1}^{-1} \Phi_{t-1}^T \eta_{t-1} y_{t-1}, \tag{4}$$

where $D_{t-1} = (\Phi_{t-1}^T \eta_{t-1} \Phi_{t-1} + \lambda I)$, $\Phi_t = [\phi(\tilde{x}_{a_\tau,\tau})^T]_{\tau=1}^{t} \in \mathbb{R}^{t \times \tilde{d}}$ and $\tilde{d}$ is the dimension of feature space $\phi(x)$. The equivalence between eqn. (2) and (4) follows from the matrix inversion lemma.

## 2   Upper Confidence Bound

**Lemma 1.** *Suppose the rewards $[r_{a_\tau,\tau}]_{\tau=1}^{T}$ are independent random variables with means $\mathbb{E}[r_{a_\tau,\tau}|\tilde{x}_{a_\tau,\tau}] = \phi(\tilde{x}_{a_\tau,\tau})^T \theta^*$, where $\|\theta^*\| \leq c$. Let $\alpha = \sqrt{\frac{\log(2TN/\delta)}{2}}$ and $\delta > 0$. With probability at least $1 - \frac{\delta}{T}$, we have that $\forall a \in [N]$*

$$|\phi(\tilde{x}_{a,t})^T \hat{\theta}_t - \phi(\tilde{x}_{a,t})^T \theta^*| \leq (\alpha + c\sqrt{\lambda}) s_{a,t},$$

*where $s_{a,t} = \sqrt{\phi(\tilde{x}_{a,t})^T D_t^{-1} \phi(\tilde{x}_{a,t})}$.*

*Proof.* Proof of this theorem is similar to proof of Lemma 1 in [4]. For simplicity we write $D_{t-1} = D, \Phi_{t-1} = \Phi, y_{t-1} = y$ and $\eta_{t-1} = \eta$. Now

$$
\begin{aligned}
\phi(\tilde{x}_{a,t})^T \hat{\theta}_t - \phi(\tilde{x}_{a,t})^T \theta^* &= \phi(\tilde{x}_{a,t})^T D^{-1} \Phi^T \eta y - \phi(\tilde{x}_{a,t})^T D^{-1} D \theta^* \\
&= \phi(\tilde{x}_{a,t})^T D^{-1} \Phi^T \eta y - \phi(\tilde{x}_{a,t})^T D^{-1} (\Phi^T \eta \Phi + \lambda I) \theta^* \\
&= \phi(\tilde{x}_{a,t})^T D^{-1} \Phi^T \eta y - \phi(\tilde{x}_{a,t})^T D^{-1} (\Phi^T \eta \Phi \theta^* + \lambda \theta^*) \\
&= \phi(\tilde{x}_{a,t})^T D^{-1} \Phi^T \eta (y - \Phi \theta^*) - \phi(\tilde{x}_{a,t})^T D^{-1} \lambda \theta^*.
\end{aligned}
$$

Therefore

$$
\begin{aligned}
|\phi(\tilde{x}_{a,t})^T \hat{\theta}_t - \phi(\tilde{x}_{a,t})^T \theta^*| &\leq |\phi(\tilde{x}_{a,t})^T D^{-1} \Phi^T \eta (y - \Phi \theta^*)| + \|\theta^*\| \|\phi(\tilde{x}_{a,t})^T D^{-1} \lambda\| \\
&\leq |\phi(\tilde{x}_{a,t})^T D^{-1} \Phi^T \eta (y - \Phi \theta^*)| + c\lambda \|\phi(\tilde{x}_{a,t})^T D^{-1}\|
\end{aligned}
$$

where the first inequality is due to Cauchy-Schwarz.

Now we know that $\mathbb{E}y = \mathbb{E}[r_{a_\tau,\tau}]_{\tau=1,...,t-1} = \Phi \theta^* \implies \mathbb{E}[y - \Phi \theta^*] = 0$. Let $f(y^1, ..., y^{t-1}) = |\phi(\tilde{x}_{a,t})^T D^{-1} \Phi^T \eta (y - \Phi \theta^*)|$ and vector $V = \phi(\tilde{x}_{a,t})^T D^{-1} \Phi^T \eta$. Then

$$|f(y^1, ...y^i, ..., y^{t-1}) - f(y^1, ...\hat{y}^i, ..., y^{t-1})| = |V_i(y^i - \hat{y}^i)| \leq |V_i|.$$

That means any component $y_i$ can change $f(y^1, ..., y^{t-1})$ by at most $|V_i|$.

Using statistical independence of all random variables $r_{a_\tau,\tau}$ in a vector $y$ and using McDiarmid's Inequality:

$$
\begin{aligned}
P(|\phi(\tilde{x}_{a,t})^T D^{-1}\Phi^T \eta(y-\Phi\theta^*)| \geq \alpha s_{a,t}) &\leq 2\exp(-\frac{2\alpha^2 s_{a,t}^2}{\|V\|^2}) \\
&\leq 2\exp(-2\alpha^2) \\
&= \frac{\delta}{TN}
\end{aligned}
$$

where the second inequality is due to

$$
\begin{aligned}
s_{a,t}^2 &= \phi(\tilde{x}_{a,t})^T D^{-1}\phi(\tilde{x}_{a,t}) \\
&= \phi(\tilde{x}_{a,t})^T D^{-1}(\Phi^T\eta\Phi + \lambda I)D^{-1}\phi(\tilde{x}_{a,t}) \\
&\geq \phi(\tilde{x}_{a,t})^T D^{-1}\Phi^T\eta\Phi D^{-1}\phi(\tilde{x}_{a,t}) \\
&\geq \phi(\tilde{x}_{a,t})^T D^{-1}\Phi^T\eta^2\Phi D^{-1}\phi(\tilde{x}_{a,t}) \\
&= \|\eta\Phi D^{-1}\phi(\tilde{x}_{a,t})\|^2 \\
&= \|V\|^2.
\end{aligned}
$$

Now applying the union bound we can see that, with probability at least $1 - \frac{\delta}{T}, \forall a \in [N]$

$$
|\phi(\tilde{x}_{a,t})^T D^{-1}\Phi^T \eta(y-\Phi\theta_a^*)| \leq \alpha s_{a,t}.
$$

Bounding the second term:

$$
\begin{aligned}
c\lambda\|\phi(\tilde{x}_{a,t})^T A_a^{-1}\| &= c\lambda\sqrt{\phi(\tilde{x}_{a,t})^T D^{-1}ID^{-1}\phi(\tilde{x}_{a,t})} \\
&\leq c\sqrt{\lambda}\sqrt{\phi(\tilde{x}_{a,t})^T D^{-1}(\lambda I + \Phi^T\Phi)D^{-1}\phi(\tilde{x}_{a,t})} \\
&= c\sqrt{\lambda}\sqrt{\phi(\tilde{x}_{a,t})^T D^{-1}\phi(\tilde{x}_{a,t})} \\
&= c\sqrt{\lambda}s_{a,t}.
\end{aligned}
$$

$\square$

We kernelize $s_{a,t}$ in the following result.

## 2.1 Proof of Lemma 1 In Main Paper

*Proof.* We use Lemma 1 to get the width and then kernelize it using techniques in [11]. Note that $\Phi\phi(\tilde{x}) = \tilde{k}_{t-1}(\tilde{x})$. When $\tilde{x} = \tilde{x}_{a,t}$, we write $\tilde{k}_{a,t} = \tilde{k}_{t-1}(\tilde{x}_{a,t})$. For simplicity we write $\eta_{t-1} = \eta$ and $\Phi_{t-1} = \Phi$. Since the matrices $(\Phi^T\eta\Phi + \lambda I), (\eta\Phi\Phi^T + \lambda I)$ are regularized, they are strictly positive definite and hence their inverses are defined. Observe that

$$(\Phi^T\eta\Phi + \lambda I)\Phi^T = \Phi^T(\eta\Phi\Phi^T + \lambda I) \tag{5}$$

by associative property of matrix multiplication and

$$\Phi^T(\eta\Phi\Phi^T + \lambda I)^{-1} = (\Phi^T\eta\Phi + \lambda I)^{-1}\Phi^T \tag{6}$$

by multiplication of $(\Phi^T\eta\Phi + \lambda I)^{-1}$ and $(\eta\Phi\Phi^T + \lambda I)^{-1}$ on both sides. Also observe that

$$(\Phi^T\eta\Phi + \lambda I)\phi(\tilde{x}_{a,t}) = (\Phi^T\eta\tilde{k}_{a,t} + \lambda\phi(\tilde{x}_{a,t}))$$

by associative property of matrix multiplication and using $\Phi\phi(\tilde{x}_{a,t}) = \tilde{k}_{a,t}$. Multiplying on the left by $(\Phi^T\eta\Phi + \lambda I)^{-1}$,

$$
\begin{aligned}
\phi(\tilde{x}_{a,t}) &= (\Phi^T\eta\Phi + \lambda I)^{-1}(\Phi^T\eta\tilde{k}_{a,t} + \lambda\phi(\tilde{x}_{a,t})) \\
&= (\Phi^T\eta\Phi + \lambda I)^{-1}\Phi^T\eta\tilde{k}_{a,t} + \lambda(\Phi^T\eta\Phi + \lambda I)^{-1}\phi(\tilde{x}_{a,t}) \\
&= \Phi^T(\eta\Phi\Phi^T + \lambda I)^{-1}\eta\tilde{k}_{a,t} + \lambda(\Phi^T\eta\Phi + \lambda I)^{-1}\phi(\tilde{x}_{a,t}) \tag{7}
\end{aligned}
$$

where the last step is due to eqn. (6).

Multiplying both sides of eqn. (7) by $\phi(\tilde{x}_{a,t})^T$ we get,

$$\phi(\tilde{x}_{a,t})^T \phi(\tilde{x}_{a,t}) \;=\; \tilde{k}_{a,t}^T (\eta \Phi \Phi^T + \lambda I)^{-1} \eta \tilde{k}_{a,t} + \lambda \phi(\tilde{x}_{a,t})^T (\Phi^T \eta \Phi + \lambda I)^{-1} \phi(\tilde{x}_{a,t})$$

or, equivalently,

$$\tilde{k}(\tilde{x}_{a,t}, \tilde{x}_{a,t}) \;=\; \tilde{k}_{a,t}^T (\eta \tilde{K}_{t-1} + \lambda I)^{-1} \eta \tilde{k}_{a,t}^T + \lambda s_{a,t}^2.$$

By rearranging terms, we get

$$s_{a,t} = \lambda^{-1/2} \sqrt{\tilde{k}(\tilde{x}_{a,t}, \tilde{x}_{a,t}) - \tilde{k}_{a,t}^T (\eta_{t-1} \tilde{K}_{t-1} + \lambda I)^{-1} \eta_{t-1} \tilde{k}_{a,t}}. \tag{8}$$

$\square$

## 3 UCB Width

In this subsection we establish a lower bound on the UCB width. To simplify the analysis we consider a problem:

$$\hat{f}_t = \operatorname*{arg\,min}_{f \in \mathcal{H}_{\tilde{k}}} \frac{1}{N} \sum_{a=1}^{N} \sum_{\tau \in t_a} (f(\tilde{x}_{a,\tau}) - r_{a,\tau})^2 + \lambda \|f\|_{\mathcal{H}_{\tilde{k}}}^2, \tag{9}$$

as $\frac{1}{n_{a,t-1}}$ obscures the analysis. In this case $s_{a,t} = \lambda^{-1/2} \sqrt{\tilde{k}(\tilde{x}_{a,t}, \tilde{x}_{a,t}) - \tilde{k}_{a,t}^T (\tilde{K}_{t-1} + \lambda I)^{-1} \tilde{k}_{a,t}}$.
Let $(\odot)$ denote the Hadamard product and $(\otimes)$ denote the Kronecker product.

**Lemma 2.** *[6] Let A be a positive definite matrix partitioned according to*

$$A = \left[ \begin{array}{c|c} A_{11} & A_{12} \\ \hline A_{21} & A_{22} \end{array} \right].$$

*Then*

$$A_{22} \geq A_{22} - A_{12}^T A_{11}^{-1} A_{12} \geq \frac{4\lambda_{\max}\lambda_{\min}}{\left(\lambda_{\max} + \lambda_{\min}\right)^2} A_{22}$$

*where $\lambda_{\max}$ and $\lambda_{\min}$ are the maximum and minimum eigenvalues of A and $A \geq B$ means $A - B$ is a positive semidefinite matrix.*

**Lemma 3.** *[8] Let $D, C$ be positive semidefinite matrices. Any eigenvalue $\lambda(D \odot C)$ of $D \odot C$ satisfies*

$$\lambda(D \odot C) \leq \lambda_{\max}(D \odot C) \leq |\max_i d_{ii}| \lambda_{max}(C)$$

*and*

$$|\min_i d_{ii}| \lambda_{min}(C) \leq \lambda_{\min}(D \odot C) \leq \lambda(D \odot C).$$

**Lemma 4.** *[5] Let $D \in \mathbb{R}^{n \times n}$ and $C \in \mathbb{R}^{m \times m}$. Any eigenvalue $\lambda(D \otimes C)$ of $D \otimes C \in \mathbb{R}^{nm \times nm}$ is equal to the product of an eigenvalue of $D$ and an eigenvalue of $C$.*

We assume that $n_{a,t} = \frac{t}{N}$ after time $t$ to get interpretibility (this is not needed for the general regret bound that we prove in Theorem 1 in main paper). For simplicity define $n_t = n_{a,t}$. Let $(\odot)$ denote the Hadamard product, $(\otimes)$ denote the Kronecker product and $\mathbb{1}_n \in R^n$ be the vector of ones. Let $K_{X_t} = [k_{\mathcal{X}}(x_{a_\tau,\tau}, x_{a_{\tau'},\tau'})]_{\tau,\tau'=1}^t$ be the $t \times t$ kernel matrix on contexts, $K_{Z_t} = [k_{\mathcal{Z}}(z_{a_\tau}, z_{a_{\tau'}})]_{\tau,\tau'=1}^t$ be the associated $t \times t$ kernel matrix based on arm similarity, and $K_Z = [k_{\mathcal{Z}}(z_a, z_a)]_{a=1}^N$ be the $N \times N$ arm similarity matrix between N arms, where $x_{a_\tau,\tau}$ is observed context and $z_{a_\tau}$ is an associated arm descriptor. Using the definition of tildek, $\tilde{k}\left((z, x), (z', x')\right) = k_{\mathcal{Z}}(z, z') k_{\mathcal{X}}(x, x')$, we can write $\tilde{K}_t = K_{Z_t} \odot K_{X_t}$. We rearrange a sequence of $x_{a_\tau,\tau}$ to get $[x_{a,\tau}]_{a=1,\tau=(t+1)_a}^N$ such that elements $(a-1)n_t$ to $an_t$ belong to arm $a$. Define $\tilde{K}_t^r, K_{X_t}^r$ and $K_{Z_t}^r$ be rearranged kernel matrices based on

the re-ordered set $[x_{a,\tau}]_{a=1,\tau=(t+1)_a}^N$. Notice that we can write $\tilde{K}_t^r = (K_Z \otimes \mathbb{1}_{n_t}\mathbb{1}_{n_t}^T) \odot K_{X_t}^r$ and the eigenvalues $\lambda(\tilde{K}_t)$ and $\lambda(\tilde{K}_t^r)$ are equal. To summarize, we have

$$\tilde{K}_t = K_{Z_t} \odot K_{X_t}$$

and

$$\lambda(\tilde{K}_t) = \lambda\Big((K_Z \otimes \mathbb{1}_{n_t}\mathbb{1}_{n_t}^T) \odot K_{X_t}^r\Big). \tag{10}$$

**Lemma 5.** *Assume $\tilde{k}(\tilde{x}, \tilde{x}) \le c_{\tilde{k}}, \forall \tilde{x} \in \tilde{X}$, and let $\tilde{K}_t$ be the final product kernel matrix and $K_Z$ be the task similarity matrix. Also write*

$$\tilde{K}_t + \lambda I_t = \left[ \begin{array}{c|c} \tilde{K}_{t-1} + \lambda I_{t-1} & \tilde{k}_{a,t} \\ \hline \tilde{k}_{a,t}^T & \tilde{k}(\tilde{x}_{a,t}, \tilde{x}_{a,t}) + \lambda \end{array} \right].$$

*Then*

$$L_{s_{a,t}} = \frac{4nc_{\tilde{k}}\lambda_{\max}(K_Z) + \lambda}{\Big(nc_{\tilde{k}}\lambda_{\max}(K_Z) + 2\lambda\Big)^2}\Big(\tilde{k}(\tilde{x}_{a,t}, \tilde{x}_{a,t}) + \lambda\Big) - 1 \le s_{a,t}^2 \le \frac{c_{\tilde{k}}}{\lambda}. \tag{11}$$

*Proof.* Using Lemma 2,

$$\tilde{k}(\tilde{x}_{a,t}, \tilde{x}_{a,t}) + \lambda - \tilde{k}_{a,t}^T(\tilde{K}_{t-1} + \lambda I_{t-1})^{-1}\tilde{k}_{a,t} \le \tilde{k}(\tilde{x}_{a,t}, \tilde{x}_{a,t}) + \lambda.$$

Subtracting $\lambda$ from both sides,

$$\lambda s_{a,t}^2 \le \tilde{k}(\tilde{x}_{a,t}, \tilde{x}_{a,t})$$

and therefore

$$s_{a,t}^2 \le \frac{c_{\tilde{k}}}{\lambda}.$$

This proves the upper bound. Again by using Lemma 2,

$$\tilde{k}(\tilde{x}_{a,t}, \tilde{x}_{a,t}) + \lambda - \tilde{k}_{a,t}^T(\tilde{K}_{t-1} + \lambda I_{t-1})^{-1}\tilde{k}_{a,t} \ge \frac{4\lambda_{\max}(\tilde{K}_t + \lambda I_t)\lambda_{\min}(\tilde{K}_t + \lambda I_t)}{\Big(\lambda_{\max}(\tilde{K}_t + \lambda I_t) + \lambda_{\min}(\tilde{K}_t + \lambda I_t)\Big)^2}\Big(\tilde{k}(\tilde{x}_{a,t}, \tilde{x}_{a,t}) + \lambda\Big)$$

Notice that the right hand side of the above equation is a monotonically decreasing function of $\frac{\lambda_{\max}}{\lambda_{\min}}$. Then

$$\begin{aligned}
\lambda s_{a,t}^2 + \lambda &\ge \frac{4\lambda_{\max}(\tilde{K}_t + \lambda I_t)\lambda_{\min}(\tilde{K}_t + \lambda I_t)}{\Big(\lambda_{\max}(\tilde{K}_t + \lambda I_t) + \lambda_{\min}(\tilde{K}_t + \lambda I_t)\Big)^2}\Big(\tilde{k}(\tilde{x}_{a,t}, \tilde{x}_{a,t}) + \lambda\Big) \\
&= \frac{\frac{4\lambda_{\max}(\tilde{K}_t) + \lambda}{\lambda_{\min}(\tilde{K}_t) + \lambda}}{\Big(\frac{\lambda_{\max}(\tilde{K}_t) + \lambda}{\lambda_{\min}(\tilde{K}_t) + \lambda} + 1\Big)^2}\Big(\tilde{k}(\tilde{x}_{a,t}, \tilde{x}_{a,t}) + \lambda\Big) \\
&= \frac{\frac{4\lambda_{\max}(\tilde{K}_t^r) + \lambda}{\lambda_{\min}(\tilde{K}_t^r) + \lambda}}{\Big(\frac{\lambda_{\max}(\tilde{K}_t^r) + \lambda}{\lambda_{\min}(\tilde{K}_t^r) + \lambda} + 1\Big)^2}\Big(\tilde{k}(\tilde{x}_{a,t}, \tilde{x}_{a,t}) + \lambda\Big) \\
&\ge \frac{\frac{4c_{\tilde{k}}\lambda_{\max}(K_{Z_t}^r) + \lambda}{\min_i K_{X_t}^r(ii)\lambda_{\min}(K_{Z_t}^r) + \lambda}}{\Big(\frac{c_{\tilde{k}}\lambda_{\max}(K_{Z_t}^r) + \lambda}{\min_i K_{X_t}^r(ii)\lambda_{\min}(K_{Z_t}^r) + \lambda} + 1\Big)^2}\Big(\tilde{k}(\tilde{x}_{a,t}, \tilde{x}_{a,t}) + \lambda\Big)
\end{aligned}$$

where $K_{X_t}^r(ii)$ are the diagonal elements of $K_{X_t}^r$ and the last inequality is due to Lemma 3. The smallest eigenvalue of $\mathbb{1}_{n_t}\mathbb{1}_{n_t}^T$ is zero and therefore according to Lemma 4, the smallest eigenvalue of $K_{Z_t}^r$ is zero. This implies

$$\lambda s_{a,t}^2 + \lambda \;\geq\; \frac{\frac{4nc_{\tilde{k}}\lambda_{\max}(K_{Z_t}^r)+\lambda}{\lambda}}{\left(\frac{nc_{\tilde{k}}\lambda_{\max}(K_{Z_t}^r)+\lambda}{\lambda}+1\right)^2}\Big(\tilde{k}(\tilde{x}_{a,t},\tilde{x}_{a,t})+\lambda\Big)$$

$$=\; \frac{4nc_{\tilde{k}}\lambda_{\max}(K_Z)+\lambda}{\Big(nc_{\tilde{k}}\lambda_{\max}(K_Z)+2\lambda\Big)^2}\Big(\tilde{k}(\tilde{x}_{a,t},\tilde{x}_{a,t})+\lambda\Big)\lambda$$

where the last equality is again due to Lemma 4. Dividing both sides by $\lambda$ and then subtracting one gives

$$s_{a,t}^2 \;\geq\; \frac{4nc_{\tilde{k}}\lambda_{\max}(K_Z)+\lambda}{\Big(nc_{\tilde{k}}\lambda_{\max}(K_Z)+2\lambda\Big)^2}\Big(\tilde{k}(\tilde{x}_{a,t},\tilde{x}_{a,t})+\lambda\Big)-1$$

$\square$

Theorem 1 below says that the lower bound on width decreases as task similarity increases. In particular, assume that all distinct tasks are similar to each other with task similarity equal to $\mu$ and there are $N$ tasks (arms). Thus $K_Z(\mu) := (1-\mu)I_N + \mu\mathbb{1}_N\mathbb{1}_N^T$.

Define

$$L_{s_{a,t}}(\mu) := \frac{4nc_{\tilde{k}}\lambda_{\max}(K_Z(\mu))+\lambda}{\Big(nc_{\tilde{k}}\lambda_{\max}(K_Z(\mu))+2\lambda\Big)^2}\Big(\tilde{k}(\tilde{x}_{a,t},\tilde{x}_{a,t})+\lambda\Big)-1.$$

**Theorem 1.** *Let $L_{s_{a,t}}$ be the lower bound on width as defined in Lemma 5. If $\mu_1 \leq \mu_2$ then*

$$L_{s_{a,t}}(\mu_1) \geq L_{s_{a,t}}(\mu_2). \tag{12}$$

*Proof.* The eigenvalues of $K_Z(\mu) = (1-\mu)I_N + \mu\mathbb{1}_N\mathbb{1}_N^T$ are $1+\mu(N-1)$ with multiplicity 1 and $1-\mu$ with multiplicity $N-1$.

That means $\lambda_{\max}(K_Z(\mu))$ is highest when tasks are more similar and it decreases as task similarity $\mu$ goes to zero. The theorem follows as $L_{s_{a,t}(\mu)}$ is a monotonically decreasing function of $\lambda_{\max}(K_Z(\mu))$
$\square$

This is important because if the lower bound on $s_{a,t}$ is small then we may be more confident about the reward estimates and this may lead to a tighter regret bound. In the next subsection we discuss the upper bound on regret.

## 4 Regret Analysis

---
**Algorithm 1** BaseKMTL-UCB at step $t$

---
1: **Input:** $\alpha \in R_+, c, \lambda, \Psi \subseteq \{1, 2, ..., t-1\}$
2: Get $\tilde{K}_\Psi = \Phi_\Psi\Phi_\Psi^T$, where $\Phi_\Psi = [\phi(\tilde{x}_{a_\tau,\tau})^T]_{\tau\in\Psi}$
3: Get $y_\Psi = \Big[r_{a_\tau,\tau}\Big]_{\tau\in\Psi}$
4: Observe context features at time $t$: $x_{a,t}$ for each $a \in N$
5: Calculate $\tilde{k}_{a,\Psi} = \Phi_\Psi^T\phi(\tilde{x}_{a,t})$ and $\tilde{k}(\tilde{x}_{a,t},\tilde{x}_{a,t})$ for each $a \in N$.
6: **for** all $a$ at time $t$ **do**
7: $\quad s_{a,t} = \lambda^{-1/2}\sqrt{\tilde{k}(\tilde{x}_{a,t},\tilde{x}_{a,t}) - \tilde{k}_{a,\Psi}^T(\tilde{K}_\Psi+\lambda I)^{-1}\tilde{k}_{a,\Psi}}$
8: $\quad ucb_{a,t} \leftarrow \tilde{k}_{a,\Psi}^T(\tilde{K}_\Psi+\lambda I)^{-1}y_\Psi + (\alpha+c\sqrt{\lambda})s_{a,t}$
9: **end for**

---

We use the Lemma 6to prove the Lemma 7

**Algorithm 2** SupKMTL-UCB

Using same notation as in [4]:

1: **Input:** $\alpha \in R_+, T \in \mathbb{N}$
2: $Q \leftarrow \lceil \log T \rceil$
3: $\Psi_1^q \leftarrow \emptyset$ and $\forall q \in [Q]$.
4: **for** $t = 1, ..., T$ **do**
5:     $q \leftarrow 1$ and $\hat{A}_1 \leftarrow [N]$
6:     **repeat**
7:        $s_{a,t}, ucb_{a,t} \leftarrow$ BaseKMTL-UCB with $\Psi_t^q$ and $\alpha$, for all $a \in \hat{A}_q$
8:        $w_{a,t} = (\alpha + c\sqrt{\lambda})s_{a,t}$
9:        **if** $w_{a,t} \leq \frac{1}{\sqrt{T}}$ for all $a \in \hat{A}_q$ **then**
10:           Choose $a_t = \text{argmax}_{a \in \hat{A}_q} ucb_{a,t}$
11:           $\Psi_{t+1}^{q'} \leftarrow \Psi_t^{q'}$ for all $q' \in [Q]$
12:        **else if** $w_{a,t} \leq 2^{-q}$ for all $a \in \hat{A}_q$ **then**
13:           $\hat{A}_{q+1} \leftarrow \{a \in \hat{A}_q | ucb_{a,t} \geq \max_{a' \in \hat{A}_q} ucb_{a',t} - 2^{1-q}\}$
14:           $q \leftarrow q + 1$
15:        **else**
16:           Choose $a_t \in \hat{A}_q$ such that $w_{a_t,t} > 2^{-q}$
17:           Update $\Psi_{t+1}^q \leftarrow \Psi_t^q \cup \{t\}$ and $\forall q' \neq q, \Psi_{t+1}^{q'} \leftarrow \Psi_t^{q'}$
18:        **end if**
19:     **until** $a_t$ is found
20:     Observe reward $r_{a_t,t}$
21: **end for**

**Lemma 6** (Lemma 1.1 in [12]). *Let $A \in \mathbb{R}^{n \times n}$ be a positive definite matrix partitioned according to*

$$A = \left[ \begin{array}{c|c} A_{11} & A_{12} \\ \hline A_{12}^T & A_{22} \end{array} \right].$$

*where $A_{11} \in \mathbb{R}^{(n-1) \times (n-1)}, A_{12} \in \mathbb{R}^{(n-1)}$ and $A_{22} \in \mathbb{R}^1$. Then $\det(A) = \det(A_{11})(A_{22} - A_{12}^T A_{11}^{-1} A_{12})$.*

Using the notations of BaseKMTL-UCB, we write $\tilde{K}_\Psi = \Phi_\Psi \Phi_\Psi^T$ and $\tilde{k}_{a,\Psi} = \Phi_\Psi^T \phi(\tilde{x}_{a,t})$ where $\Phi_\Psi = [\phi(\tilde{x})_{a_\tau,\tau}^T]_{\tau \in \Psi}$ and $\Psi \subseteq \{1, ..., t-1\}$. Define

$$\tilde{K}_{\Psi+1} + \lambda I = \left[ \begin{array}{c|c} \tilde{K}_\Psi + \lambda I_{|\Psi|} & \tilde{k}_{a,\Psi} \\ \hline \tilde{k}_{a,\Psi}^T & \tilde{k}(\tilde{x}_{a,t}, \tilde{x}_{a,t}) + \lambda \end{array} \right]$$

Also, define $\tilde{k}_1 = \tilde{k}(\tilde{x}_{a_\sigma,\sigma}, \tilde{x}_{a_\sigma,\sigma})$, where $\sigma$ is the smallest element of $\Psi$.

**Lemma 7.** *Using notations in BaseKMTL-UCB and suppose $|\Psi| \geq 2$. Then*

$$\sum_{\tau \in \Psi} s_{a_\tau,\tau}^2 \leq 2m \log g(\Psi),$$

*where $m = \max(1, \frac{c_{\tilde{k}}}{\lambda})$ and*

$$g(\Psi) = \frac{\det(\tilde{K}_{\Psi+1} + \lambda I)}{\lambda^{|\Psi|+1}}.$$

*Proof.* Using the Lemma 6,

$$
\begin{aligned}
\det(\tilde{K}_{\Psi+1} + \lambda I) &= (\tilde{k}_1 + \lambda) \prod_{\tau \in \Psi \backslash \{\sigma\}} \lambda(1 + s_{a_\tau,\tau}^2) \\
&= \lambda(\frac{\tilde{k}_1}{\lambda} + 1) \prod_{\tau \in \Psi \backslash \{\sigma\}} \lambda(1 + s_{a_\tau,\tau}^2) \\
&= \lambda \prod_{\tau \in \Psi} \lambda(1 + s_{a_\tau,\tau}^2),
\end{aligned}
$$

where the last step is because $s_{a_\sigma,\sigma}^2 = \frac{k_1}{\lambda}$.

From Lemma 5, $\max s_{a_\tau,\tau}^2 = \frac{c_{\tilde{k}}}{\lambda}$. When $\frac{c_{\tilde{k}}}{\lambda} \leq 1$, using $x \leq 2\log(1 + x), \forall x \in [0,1]$, $s_{a_\tau,\tau}^2 \leq 2\log(1 + s_{a_\tau,\tau}^2)$. In this case,

$$
\begin{aligned}
\sum_{\tau \in \Psi} s_{a_\tau,\tau}^2 &\leq 2 \sum_{\tau \in \Psi} \log(1 + s_{a_\tau,\tau}^2) \\
&= 2\log \prod_{\tau \in \Psi} (1 + s_{a_\tau,\tau}^2) \\
&= 2\log \frac{\det(\tilde{K}_{\Psi+1} + \lambda I)}{\lambda^{|\Psi|+1}}.
\end{aligned}
$$

When $\frac{c_{\tilde{k}}}{\lambda} > 1$,

$$
\begin{aligned}
\sum_{\tau \in \Psi} \frac{c_{\tilde{k}}}{\lambda} \frac{\lambda}{c_{\tilde{k}}} s_{a_\tau,\tau}^2 &\leq \frac{2c_{\tilde{k}}}{\lambda} \sum_{\tau \in \Psi} \log(1 + \frac{\lambda}{c_{\tilde{k}}} s_{a_\tau,\tau}^2) \\
&\leq \frac{2c_{\tilde{k}}}{\lambda} \sum_{\tau \in \Psi} \log(1 + s_{a_\tau,\tau}^2) \\
&= \frac{2c_{\tilde{k}}}{\lambda} \log \prod_{\tau \in \Psi} (1 + s_{a_\tau,\tau}^2) \\
&= \frac{2c_{\tilde{k}}}{\lambda} \log \frac{\det(\tilde{K}_{\Psi+1} + \lambda I)}{\lambda^{|\Psi|+1}}.
\end{aligned}
$$

Combining both cases,

$$
\begin{aligned}
\sum_{\tau \in \Psi} s_{a_\tau,\tau}^2 &\leq 2 \max(1, \frac{c_{\tilde{k}}}{\lambda}) \log \frac{\det(\tilde{K}_{\Psi+1} + \lambda I)}{\lambda^{|\Psi|+1}} \\
&= 2m \log g(\Psi).
\end{aligned}
$$

$\square$

**Lemma 8.** *Using the same notations as in Lemma 7,*

$$
\sum_{\tau \in \Psi} s_{a_\tau,\tau} \leq \sqrt{2m|\Psi| \log g(\Psi)}
$$

*Proof.*

$$
\begin{aligned}
\sum_{t \in \Psi} s_{a_\tau,\tau} &\leq \sqrt{|\Psi| \sum_{\tau \in \Psi} s_{a_\tau,\tau}^2} \\
&\leq \sqrt{2|\Psi|m \log \frac{\det(\tilde{K}_{\Psi+1} + \lambda I)}{\lambda^{|\Psi|+1}}}
\end{aligned}
$$

where the first inequality is due to Cauchy-Schwarz and the last inequality is due to Lemma 7. $\square$

**Lemma 9.** *[1] Using notations in SupKMTL-UCB, for each $t \in [T]$, $q \in [Q]$, and any fixed sequence of feature vectors $x_{a_t,t}$ with $t \in \Psi_t^q$, the corresponding rewards $r_{a_t,t}$ are independent random variables such that $\mathbb{E}[r_{a_t,t}] = \phi(\tilde{x}_{a_t,t})^T \theta^*$.*

**Lemma 10.** *[1] Using notations in SupKMTL-UCB, let $\|\theta^*\| \le c$ and $a_t^*$ be the best arm at time $t$. With probability $1 - \delta Q$ and $\forall t \in [T], q \in [Q]$, the following hold*

- $|\phi(\tilde{x}_{a,t})^T \hat{\theta}_t - \mathbb{E}[r_{a,t}|x_{a,t}]| \le \left( \sqrt{\frac{\log 2TN/\delta}{2}} + \sqrt{\lambda} c \right) s_{a,t}$

- $a_t^* \in \hat{A}_q$

- $\mathbb{E}[r_{a_t^*,t}] - \mathbb{E}[r_{a,t}] \le 2^{3-q}$.

**Lemma 11.** *Using notations in SupKMTL-UCB, $\forall q \in [Q]$,*

$$|\Psi_{T+1}^q| \le 2^q \left( \sqrt{\frac{\log 2TN/\delta}{2}} + c\sqrt{\lambda} \right) \sqrt{2m \left( \log g([T]) \right) |\Psi_{T+1}^q|}$$

*where $[T] = \{1, ..., T\}$.*

*Proof.*

$$
\begin{aligned}
\sum_{t \in \Psi_{T+1}^q} w_{a_t,t} &= \sum_{t \in \Psi_{T+1}^q} \left( \sqrt{\frac{\log 2TN/\delta}{2}} + c\sqrt{\lambda} \right) s_{a_t,t} \\
&\le \left( \sqrt{\frac{\log 2TN/\delta}{2}} + c\sqrt{\lambda} \right) \sqrt{2m |\Psi_{T+1}^q| \log g(\Psi_{T+1}^q)} \\
&\le \left( \sqrt{\frac{\log 2TN/\delta}{2}} + c\sqrt{\lambda} \right) \sqrt{2m \left( \log g([T]) \right) |\Psi_{T+1}^q|}
\end{aligned}
$$

where the first inequality is due to Lemma 8 and the last inequality holds because $1 + s_{a_t,t}^2 \ge 1$ for all $t$.

From the third step (line 16) in SupKMTL-UCB algorithm 2, we choose and alternative $a_t \in \hat{A}_q$ such that $w_{a_t,t} \ge 2^{-q}$ and include that $t$ in $\Psi_{t+1}^q$ for the next round of estimates. Therefore,

$$\sum_{t \in \Psi_{T+1}^q} w_{a_t,t} \ge 2^{-q} |\Psi_{T+1}^q|$$

.

Combining the above two equations completes the proof. $\square$

**Lemma 12.** *[Azuma's inequality [2]] Let $r_1, ..., r_T$ be random variables with $|r_\tau| \le a_\tau$, for some $a_1, ..., a_T \ge = 0$. Then*

$$P \left( \left| \sum_{\tau=1}^T r_\tau - \sum_{\tau=1}^T \mathbb{E}[r_\tau | r_1, ..., r_{\tau-1}] \right| \ge B \right) \le 2 \exp \left( -\frac{B^2}{2 \sum_{\tau=1}^T a_\tau^2} \right) \tag{13}$$

## 4.1 Proof of Theorem 1 in Main paper

We use same proof technique proposed by Auer et al. [1].

*Proof.* Let $\Psi_0$ be the set of trials for which an alternative ($w_{a,t} \le \frac{1}{\sqrt{T}}$) at line 9 of SupKMTL-UCB algorithm 2 is chosen . Since $2^{-Q} \le \frac{1}{\sqrt{T}}$, we have $\{1, ..., T\} = \Psi_0 \cup \bigcup_q \Psi_{T+1}^q$.

With probability $1 - \delta Q$,

$$
\begin{aligned}
\mathbb{E}[R(T)] &= \sum_{t=1}^{T} \mathbb{E}[r_{a_t^*,t}] - \mathbb{E}[r_{a_t,t}] \\
&= \sum_{t \in \Psi_0} \mathbb{E}[r_{a_t^*,t}] - \mathbb{E}[r_{a_t,t}] + \sum_{q=1}^{Q} \sum_{t \in \Psi_{T+1}^q} \mathbb{E}[r_{a_t^*,t}] - \mathbb{E}[r_{a_t,t}] \\
&\leq \frac{2}{\sqrt{T}} \Psi_0 + \sum_{q=1}^{Q} \sum_{t \in \Psi_{T+1}^q} \mathbb{E}[r_{a_t^*,t}] - \mathbb{E}[r_{a_t,t}] \\
&\leq \frac{2}{\sqrt{T}} T + \sum_{q=1}^{Q} \sum_{t \in \Psi_{T+1}^q} 2^{3-q} \\
&\leq 2\sqrt{T} + \sum_{q=1}^{Q} 2^{3-q} |\Psi_{T+1}^q| \\
&\leq 2\sqrt{T} + \sum_{q=1}^{Q} 2^{3-q} 2^q \left( \sqrt{\frac{\log 2TN/\delta}{2}} + c\sqrt{\lambda} \right) \sqrt{2m\left(\log g([T])\right) |\Psi_{T+1}^q|} \\
&\leq 2\sqrt{T} + 8 \left( \sqrt{\frac{\log 2TN/\delta}{2}} + c\sqrt{\lambda} \right) \sqrt{2m\left(\log g([T])\right)} \sum_{q=1}^{Q} \sqrt{|\Psi_{T+1}^q|} \\
&\leq 2\sqrt{T} + 8 \left( \sqrt{\frac{\log 2TN/\delta}{2}} + c\sqrt{\lambda} \right) \sqrt{2m\left(\log g([T])\right)} \sqrt{Q \sum_{q=1}^{Q} |\Psi_{T+1}^q|} \\
&\leq 2\sqrt{T} + 8 \left( \sqrt{\frac{\log 2TN/\delta}{2}} + c\sqrt{\lambda} \right) \sqrt{2m\left(\log g([T])\right)} \sqrt{QT}
\end{aligned}
$$

where the first inequality is because of line 9 of SupKMTL-UCB algorithm 2, the second inequality is due to Lemma 10 and the fourth inequality is due to Lemma 11.

Using $B = \sqrt{2T \log(2/\delta)}$ and $a_\tau = 1$ in Azuma's inequality (Lemma 12), with probability at least $1 - \delta(Q+1)$,

$$
\begin{aligned}
R(T) &\leq \mathbb{E}[R(T)] + \sqrt{2T \log(2/\delta)} \\
&\leq 2\sqrt{T} + 8 \left( \sqrt{\frac{\log 2TN/\delta}{2}} + c\sqrt{\lambda} \right) \sqrt{2m\left(\log g([T])\right)} \sqrt{QT} + \sqrt{2T \log(2/\delta)} \\
&\leq 2\sqrt{T} + 10 \left( \sqrt{\frac{\log 2TN/\delta}{2}} + c\sqrt{\lambda} \right) \sqrt{2m\left(\log g([T])\right)} \sqrt{QT}.
\end{aligned}
$$

Replacing $\delta$ with $\frac{\delta}{Q+1}$, we get that with probability at least $1 - \delta$,

$$
\begin{aligned}
R(T) &\leq 2\sqrt{T} + 10 \left( \sqrt{\frac{\log 2TN(Q+1)/\delta}{2}} + c\sqrt{\lambda} \right) \sqrt{2m\left(\log g([T])\right)} \sqrt{QT} \qquad (14) \\
&\leq 2\sqrt{T} + 10 \left( \sqrt{\frac{\log 2TN(\log(T)+1)/\delta}{2}} + c\sqrt{\lambda} \right) \sqrt{2m \log g([T])} \sqrt{T \lceil \log(T) \rceil} \quad (15)
\end{aligned}
$$

$\square$

We use following definitions and lemmas to interpret the regret bound and to establish a regret bound in terms of the effective rank of the kernel matrix.

**Definition 1.** *Let $x, y \in \mathbb{R}^n$ and $x_1 \geq x_2 \geq .... \geq x_n$, $y_1 \geq y_2 \geq .... \geq y_n$ . We say $x$ is majorized by $y$, i.e. $x \prec y$, if $\sum_{i=1}^{k} x_i \leq \sum_{i=1}^{k} y_i$, for $k = 1, ..., n-1$ and $\sum_{i=1}^{n} x_i = \sum_{i=1}^{n} y_i$.*

**Definition 2.** *A real valued function on $g$ defined on set $\mathcal{S} \subset \mathbb{R}^n$ is said to be Schur concave on $\mathcal{S}$ if $x \prec y \implies g(x) \geq g(y)$.*

**Lemma 13.** *[7] If $x, y \in \mathbb{R}^n_+$ and $x \prec y$, then $\prod_{i=1}^{n} x_i \geq \prod_{i=1}^{n} y_i$. This means $\prod x_i$ is a Schur concave function.*

**Lemma 14.** *[3] Let $A, B$ be positive semidefinite matrices of the same size and let all elements on diagonal of $B$ are 1. Then $\lambda(A \odot B) \prec \lambda(A)$.*

**Lemma 15.** *[5] Let $A, B$ be matrices of size $\mathbb{R}^{n \times m}$ then $\mathrm{rank}(A \odot B) \leq \mathrm{rank}(A)\,\mathrm{rank}(B)$.*

**Lemma 16.** *[ Arithmetic Mean-Geometric Mean Inequality [9]] For every sequence of nonnegative real numbers $a_1, a_2, ...a_n$ one has*

$$\left(\prod_{i=1}^{n} a_i\right)^{1/n} \leq \frac{\sum_{i=1}^{n} a_i}{n}$$

*with equality if and only if $a_1 = a_2 = ... = a_n$.*

### 4.2 Proof of Theorem 2 in Main Paper

Suppose the rank of $\tilde{K}_{T+1}$ is $r$. Hence only the first $r$ eigenvalues are non zero. In that case $g([T])$ attains its maximum when each of these $r$ eigenvalues is equal to $\frac{\mathrm{trace}(\tilde{K}_{T+1})}{r}$ (using Lemma 16). Thus,

$$
\begin{aligned}
g([T]) &= \frac{\prod_{i=1}^{T+1}(\lambda_i + \lambda)}{\lambda^{T+1}} \\
&\leq \frac{\prod_{i=1}^{r}(\mathrm{trace}(\tilde{K}_{T+1})/r + \lambda)}{\lambda^r} \\
&= \left(\frac{\mathrm{trace}(\tilde{K}_{T+1})/r + \lambda}{\lambda}\right)^r.
\end{aligned}
$$

It follows that,

$$
\begin{aligned}
\log(g([T])) &\leq r \log\left(\frac{\mathrm{trace}(\tilde{K}_{T+1})/r + \lambda}{\lambda}\right) \\
&\leq r \log\left(\frac{\mathrm{trace}(\tilde{K}_{T+1}) + \lambda}{\lambda}\right) \\
&= r_z r_x \log\left(\frac{\mathrm{trace}(\tilde{K}_{T+1}) + \lambda}{\lambda}\right) \\
&\leq r_z r_x \log\left(\frac{(T+1)c_{\tilde{k}} + \lambda}{\lambda}\right),
\end{aligned}
$$

where the second inequality is due to Lemma 15.

### 4.3 Proof of Theorem 3 in Main Paper

*Proof.* Suppose the $\tilde{K}_{T+1}(\mu_1)$ and $\tilde{K}_{T+1}(\mu_2)$ are final kernel matrices after time $T$, $\tilde{K}_{T+1}^r(\mu_1)$ and $\tilde{K}_{T+1}^r(\mu_2)$ are corresponding matrices using the definition 10. Also suppose that $K_Z(\mu_1)$ and $K_Z(\mu_2)$ are task similarity matrices.

$$
\begin{aligned}
\tilde{K}_{T+1}^r(\mu_1) &= (K_Z(\mu_1) \otimes \mathbb{1}_{n_{T+1}} \mathbb{1}_{n_{T+1}}^T) \odot K_{X_{T+1}}^r \\
&= (K_Z(\frac{\mu_1}{\mu_2}) \otimes \mathbb{1}_{n_{T+1}} \mathbb{1}_{n_{T+1}}^T) \odot (K_Z(\mu_2) \otimes \mathbb{1}_{n_{T+1}} \mathbb{1}_{n_{T+1}}^T) \odot K_{X_{T+1}}^r
\end{aligned}
$$

and using the Lemma 14, we have

$$\lambda(\tilde{K}_{T+1}(\mu_1)) \prec \lambda(\tilde{K}_{T+1}(\mu_2))$$

This implies

$$\lambda(\tilde{K}_{T+1}(\mu_1)) + \lambda \prec \lambda(\tilde{K}_{T+1}(\mu_2)) + \lambda.$$

Using the Lemma 13, we conclude that

$$\prod_{t=1}^{T+1}(\lambda_t(\tilde{K}_{T+1}(\mu_1)) + \lambda) \geq \prod_{t=1}^{T+1}(\lambda_t(\tilde{K}_{T+1}(\mu_2)) + \lambda).$$

This completes the proof. □

### 4.4 Proof of Corollary 1

*Proof.* Let's find the upper bound of maximum of $g([T])$. We know that $r\lambda\log T \geq \sum_{i=r+1}^{T+1}\lambda_i$. Let $\epsilon$ be a constant such that $r\lambda\log T = \sum_{i=r+1}^{T+1}\lambda_i + \epsilon$. Notice that $\epsilon \leq (T+1)c_{\tilde{k}}$. Consider

$$\begin{aligned}
\max \quad & \prod_{i=1}^{T+1} \quad (\lambda_i + \lambda) \\
\text{s.t.} \quad & \sum_{i=1}^{r} \quad \lambda_i + \lambda = (T+1)c_{\tilde{k}} + r\lambda - r\lambda\log T + \epsilon \\
\text{and} \quad & \sum_{i=r+1}^{T+1} \quad \lambda_i + \lambda = r\lambda\log T - \epsilon + (T+1-r)\lambda
\end{aligned}$$

Using Lemma 16, the maximum of above constrained optimization problem occurs at

$$\lambda_i + \lambda = \begin{cases} \frac{(T+1)c_{\tilde{k}} + r\lambda - r\lambda\log T + \epsilon}{r}, & \text{if } \lambda_i \leq r, \\ \frac{r\lambda\log T + (T+1-r)\lambda}{(T+1-r)} - \frac{\epsilon}{T+1-r} & \text{otherwise.} \end{cases} \tag{16}$$

Therefore,

$$\begin{aligned}
g([T]) &= \prod_{t=1}^{T+1} \frac{(\lambda_t + \lambda)}{\lambda} \\
&\leq \left(\frac{(T+1)c_{\tilde{k}} + r\lambda - r\lambda\log T + \epsilon}{r\lambda}\right)^r \left(\frac{r\lambda\log T + (T+1-r)\lambda}{(T+1-r)\lambda}\right)^{T+1-r} \\
&= \left(\frac{(T+1)c_{\tilde{k}} + r\lambda - r\lambda\log T + \epsilon}{r\lambda}\right)^r \left(\frac{r\log T + (T+1-r)}{(T+1-r)}\right)^{T+1-r} \\
&= \left(\frac{(T+1)c_{\tilde{k}} + r\lambda - r\lambda\log T + \epsilon}{r\lambda}\right)^r \left(\frac{r\log T}{T+1-r} + 1\right)^{T+1-r} \\
&= \left(\frac{(T+1)c_{\tilde{k}} + r\lambda - r\lambda\log T + \epsilon}{r\lambda}\right)^r \left(\frac{r\log T}{T+1-r} + 1\right)^{T+1-r} \\
&\leq \left(\frac{(T+1)c_{\tilde{k}} + r\lambda - r\lambda\log T + \epsilon}{r\lambda}\right)^r \left(\frac{r\log(T+r-1)}{T} + 1\right)^{T} \\
&\leq \left(\frac{(T+1)c_{\tilde{k}} + r\lambda - r\lambda\log T + \epsilon}{r\lambda}\right)^r \exp\left(r\log(T+r-1)\right)
\end{aligned}$$

where the first inequality is due to eqn. (16), the second inequality holds because $(1 + \frac{\log(x)}{x})^x$ is monotonically increasing function $\forall x \geq 1$ and the last inequality holds because $\log(1+x) \leq x, \forall x > -1$.

Taking log on both sides

$$
\begin{aligned}
\log(g([T])) &\leq r\log\left(\frac{(T+1)c_{\tilde{k}} + r\lambda - r\lambda\log T + \epsilon}{r\lambda}\right) + r\log(T + r - 1) \\
&\leq r\log\left(\frac{(T+1)c_{\tilde{k}} + r\lambda - r\lambda\log T + \epsilon}{r\lambda}\right) + r\log(2T) \\
\log(g([T])) &\leq r\log\left(2T\frac{2(T+1)c_{\tilde{k}} + r\lambda - r\lambda\log T}{r\lambda}\right).
\end{aligned}
$$

$\square$

## 5   Comment on Scalability

As number of time steps $t$ increases, kernel matrix $\tilde{K}_t$ grows in size and that has an effect on how well one can scale the proposed algorithm. While scalability was not our focus, there are at least a couple of obvious ways to address this issue. One would be rank one updates of the kernel matrix, and another would be kernel approximation techniques such as the Nystrom method. Both are straightforward to implement. We had implemented the latter, and noticed that such an approximation has a more detrimental effect on Kernel-UCB than our method. One possible reason could be that the Nystrom method assumes the kernel matrix has low rank, and when there is more task similarity, the rank of our kernel matrix is lower compared to that of Kernel-UCB.

## 6   Results

We run every experiment 10 times and calculate empirical mean regret. Confidence interval is calculated using standard deviation (mean $\pm$ standard deviation) of results of these 10 experiments.

Figure 1: Results on Multiclass Dataset with confidence interval