[Reviews · NeurIPS 2017]

Reviewer 1



Summary. The paper is about contextual bandits with N arms. In each round the learner observes a context x_{ti} for each arm, chooses an arm to pull and receives reward r_{ti}. The question is what structure to impose on the rewards. The authors note that E[r_{ti}] = < x_{ti}, theta > is a common choice, as is E[r_{ti}] = < x_{ti}, theta_i > The former allows for faster learning, but has less capacity while the latter has more capacity and slower learning. The natural question addressed in this paper concerns the middle ground, which is simultaneously generalized by kernelization. The main idea is to augment the context space so the learner observes (z_{ti}, x_{ti}) where z_{ti} lies in some other space Z. Then a kernel can be defined on this augmented space that measures similarity between contexts and determines the degree of sharing between the arms. Contribution. The main contribution as far as I can tell is the idea to augment the context space in this way. The regret analysis employs the usual techniques. Novelty. There is something new here, but not much. Unless I am somehow mistaken the analysis of Valko et al. should apply directly to the augmented contexts with more-or-less the same guarantees. So the new idea is really the augmentation. Impact. It's hard to tell what will be the impact of this paper. From a theoretical perspective there is not much new. The practical experiments are definitely appreciated, but not overwhelming. Eg., what about linear Thompson sampling? Correctness. I only skimmed the proofs in the supplementary material, but the bound passes plausibility tests. Overall. This seems like a borderline paper to me. I would increase my score if the authors can argue convincingly that the theoretical results are really doing more than the analysis in the cited paper of Valko et al. Other comments. - On L186 you remark that under "further assumption that after time t, n_{a,t} = t/N". But this assumption is completely unjustified, so how meaningful are conclusions drawn from it? - It's a pity that the analysis was not possible for Algorithm 1. I presume the sup-"blah blah" algorithms don't really work? It could be useful to show their regret relative to Algorithm 1 in one of the figures. Someone needs to address these issues at some point (but I know this is a tricky problem). Minors. - I would capitalize A_t and other random variables. - L49: "one estimate" -> "one estimates". - L98: "z_a" -> "z_a \in \mathcal Z". - L110: The notation t_a as a set that depends on t and a is just odd. - L114: Why all the primes? - There must be some noise assumption on the rewards (or bounded, or whatever). Did I miss it? - L135: augmented context here is also (a, x_a).

Reviewer 2



This paper introduces a multitask bandit learning approach. It takes after two existing contributions: Valko et al. 2013 on kernelised contextual bandits, Evgueniou and Pontil, 2004 on regularized multitask learning. The authors of the present paper provide a way to estimate the similarities between the tasks if it is not given, which is essential for real-world data. Pros of the paper: - the problem of learning multitask contextual bandits is of importance for many practical problems (e.g. recommendation); - the mathematical anaylsis, as far as I have checked, is correct; - results from numerical simulations are convincing. Cons: - I would point out that the paper provides an incremental contribution and/or that the novelty is not well sold. For instance, it seems a lot of the work provided here is similar to the work of Valko et al 2013. What if that work is implemented with multitask kernels ? Would the resulting algorithm be very different from that proposed in the present paper ? - there is the question of the computational complexity induced by the growing kernel matrix K_{t-1}: something should be said here. - there is the frustrating proof fo the regret that mentions two algorithms SupKMTL-UCB and BaseKMTL-UCB that are only given in the supplementary material: the authors should at least provide the main lines of these algorithms in the main text. Otherwise, Theorem 4.1 cannot be understood. All in all, the paper addresses an interesting problem. However, there is some drawbacks regarding 1) the incrementality of the contribution, 2) some algorithmic points (e.g. growing kernel matrix) and 3) the presentation of Theorem 4.1.